# Assessment of sediment porewater toxicity in Biscayne National Park with sea urchin (*Lytechinus variegatus*) embryos

Lisa A. May[1]*, Elizabeth Murphy McDonald[1], Ronald T. Kothera[1], Catherine Anna Toline[2], Vanessa McDonough[3], Zachary J. Moffitt[1], Carl V. Miller[1], Cheryl M. Woodley[4]*

**1** Consolidated Safety Services, Inc. Under Contract to the Key Species and Bioinformatics Branch, Stressor Detection and Impacts Division, National Centers for Coastal Ocean Science, National Oceanic and Atmospheric Administration, Charleston, SC, United States of America, **2** National Park Service, Region 2, South Atlantic Gulf, Charleston, SC, United States of America, **3** National Park Service, Biscayne National Park, Homestead, FL, United States of America, **4** National Oceanic and Atmospheric Administration, National Centers for Coastal Ocean Science, Stressor Detection and Impacts Division, Key Species and Bioinformatics Branch, Charleston, SC, United States of America

* lisa.may@noaa.gov (LAM); cheryl.woodley@noaa.gov (CMW)

**Data Availability Statement:** All relevant data are within the manuscript and its Supporting Information files.

## Abstract

The sea urchin embryo development toxicity test was used to investigate toxicity of the benthic substrate in Biscayne National Park (BISC). Twenty-five sites were selected based upon a high potential for anthropogenic stressor input (e. g., hydrocarbons, personal care products, nutrients, etc.) or proximity to coral reef habitats. We found that sediment interstitial water (porewater) was toxic to urchin embryos at 22 of 25 sites. Healthy sites included two coral reefs (Anniversary Reef and Marker 14 Reef) and Turkey Point Channel. Discrete areas of BISC have highly toxic sediments and the presence of sediment contaminants could negatively impact reproduction, growth and population density of benthic invertebrates, such as corals. Results of the sea urchin embryo development toxicity test can be used as a baseline assessment for monitoring improvements or degradation in ecosystem health and could be a valuable tool to investigate the suitability of degraded habitats for future reef restoration. Since the last comprehensive environmental assessment of BISC was performed in 1999, further investigation into the sources of toxicity at BISC is needed.

## Introduction

Biscayne National Park (BISC) comprises nearly 700 square kilometers in southeast Florida and protects four distinct ecosystems: (1) terrestrial mangrove shorelines, (2) shallow estuarine system (Biscayne Bay) with diverse bottom communities, (3) keys (islands), and (4) coral reefs and hardbottom habitat [1]. The Park is located in a transition zone between two biogeographic realms (Nearctic and Neotropical), resulting in a unique diversity of plant and animal life [2]. In 2019–2020, BISC averaged over 500,000 visitors annually [3]. In addition to the recreational value it provides, BISC protects diverse species and sensitive habitats critical for maintaining balanced ecosystems. The fragile biome of the Park may be impacted by nearby

**Funding:** This work was supported by a grant to C. M. W. from the National Park Service (Project # PMIS 239736). YES - Employees of the National Park Service assisted with study design and preparation of the manuscript.

**Competing interests:** The authors have declared that no competing interests exist.

land use, which includes power stations at Turkey Point (oil-fired, oil/natural gas and nuclear), agriculture, the South Miami-Dade landfill, a sewage treatment facility at Black Point, a large motor speedway and a significant urban population in Miami-Dade County that has increased 66% since the Park was founded [4].

Resource managers are faced with an ever-increasing number and magnitude of threats from water-based tourism and recreation, coastal development, and co-location with major urban centers and shipping ports [5–7]. In 2015, NOAA, working with BISC and other stakeholders, selected BISC as a Habitat Focus Area under NOAA's Habitat Blueprint Program. The selection was based on evidence of degraded water quality from increased nutrients (nitrogen and phosphorus) that resulted in algal overgrowths (a symptom of eutrophication). Eutrophication in South Florida usually results from two sources: input of human sewage (from outfalls or leaky septic systems) and fertilizer runoff (from agriculture and/or intensive landscaping). Along with inputs of nitrogen and phosphorous, these sources also carry a diversity of chemical pollutants that can impact receiving habitats. From agriculture, this includes livestock and crop-protection products (pesticide, herbicides, fungicides, antibiotics, steroids). Human sewage releases, industrial discharges and runoff from storm events can release hundreds to thousands of chemicals into a habitat [8–10]. These chemicals (including contaminants of emerging concern, such as pharmaceuticals and personal care products) may act as endocrine disruptors and may be carcinogenic, genotoxic (damages DNA) or teratogenic (interrupts normal development). These chemical pollutants can disrupt critical physiological processes such as reproduction and early life-history development, which are necessary for maintaining a population [11].

It has long been recognized that sediments are a reservoir for many contaminants, such as hydrocarbons from runoff or antifoulant compounds from boat use. The interstitial water within sediments (porewater) is a valuable tool for investigating water quality parameters that are difficult to capture in the water column. Porewater toxicity testing using sea urchin embryo development can reveal biological impairment of the marine environment [12]. A recent (2015) NOAA National Ocean Service survey of the Florida Keys National Marine Sanctuary evaluated sediment samples from 28 sites along Florida's Coral Reef (Key Biscayne to Dry Tortugas National Park) using multiple indicators of sediment quality, including the sea urchin embryo development test (SUETOX) to determine toxicity of the sediment porewater [13]. Station 1 immediately south of Miami, FL, at the northern end of the study area and proximal to BISC, resulted in 100% toxicity to sea urchin embryo development. All measured chemical contaminants were below corresponding Effects Range-Low (ERL) and Effects Range-Median (ERM) sediment quality guidelines [14, 15] and there was no evidence of co-occurring bioeffects based on other benthic infaunal or sediment toxicity indicators. However, additional single-step fractionation of sediment porewaters showed evidence that the observed sea urchin embryo toxicity at this site may have been due, at least in part, to neutral analytes or compounds with aromatic or hydrophobic residues (e.g., pesticides, herbicides, steroid hormones and pharmaceuticals). Other unmeasured stressors or confounding natural factors could have contributed as well. Regardless of the source, results of the sea urchin assay at this site suggest the possibility of sediment toxicity in other nearby portions of BISC that were not surveyed in the prior NOAA study.

In 2018, two sediment samples were collected from Ball Buoy and Brewster reefs in BISC. The SUETOX test was performed on the extracted porewater. Again, test results indicated severe impairment with 100% toxicity for both sites (our unpublished data). Embryos scored as arrested development (e.g., cell stage, blastula, gastrula) were the most prevalent (83% for Ball Buoy and 99% for Brewster). Based on these results, an in-depth study was undertaken to more fully document potential impairment in the Park. Marine sediment samples were

collected from coral reefs and other areas of concern in August 2019. The sediment porewater was extracted from each sample and subjected to the SUETOX test. The results of the SUETOX test were used to understand the sediment quality in the Park. Results are reported herein.

## Materials and methods

### Sample collection and porewater extraction

Field sediment collections were performed under permit number BISC-2019-SCI-0020 granted by the US National Park Service. Site selection within BISC was targeted to areas of expected pollution (e.g., urbanized/industrialized areas, marinas and high-use recreational areas as observed by National Park Service personnel) and sensitive coral reef habitats (Table 1). One site (Brewster) is located proximally to Station 1 in the Balthis et al. [13] study. Marine sediments were collected by National Park Service divers from 25 sites within BISC using five 60 mL coring syringes per site [16]. The sediment from the five coring syringes from each site were pooled into a Teflon bag (7" by 6", Welch Fluorocarbon, Dover, NH), sealed with a Clip-n-Seal® closure (Welch Fluorocarbon) and held on ice for transport to the field laboratory. Porewater was extracted from the sediment using a 10 mL glass pipet and placed into a clean 50 mL Teflon PFA centrifuge tube (Savillex, Eden Prairie, MN). Samples were centrifuged for 20 min at 1200 x $g$ and the clarified samples were transferred to clean 90 mL Teflon jars (Savillex). Extracted porewater was frozen at -20°C and frozen samples were placed in a

**Table 1. Biscayne National Park sediment porewater sampling site names and GPS locations.**

| Site | Latitude | Longitude | Site Description |
|---|---|---|---|
| Brewster Reef | 25.566436 | -80.099466 | Coral reef |
| Stiltsville A-Frame | 25.654753 | -80.168390 | Near large population, high visitor use |
| Stiltsville Baldwin Sessions | 25.651981 | -80.170485 | Near large population, high visitor use |
| Stiltsville Miami Spring Power Boat Club | 25.641381 | -80.181345 | Near large population, high visitor use |
| Boca Chita Harbor | 25.524212 | -80.175167 | High boat use |
| "Boiler" safety valve | 25.583967 | -80.166783 | Large volume of tidal flushing, shallow |
| Mowry Canal | 25.470368 | -80.340192 | Popular fishing spot, very polluted |
| Soldier Key | 25.589932 | -80.161666 | Unique island, shallow |
| Lirman Nursery 1 | 25.488800 | -80.109100 | Coral nursery |
| Lirman Nursery 2 | 25.473400 | -80.128700 | Coral nursery |
| Lirman Nursery 3 | 25.362600 | -80.166400 | Coral nursery |
| University Dock | 25.479467 | -80.189688 | High visitor use |
| Anniversary Reef | 25.388250 | -80.164400 | Coral reef, snorkeling site |
| Mandalay | 25.442233 | -80.121317 | Snorkeling site |
| Alina's Reef | 25.386383 | -80.162983 | Coral reef |
| Marker 14 Reef | 25.463940 | -80.168480 | Coral reef |
| Money Reef | 25.444220 | -80.176210 | Coral Reef |
| Biscayne Boat Basin | 25.463402 | -80.334184 | Heavy boat use, Park staff and visitors |
| Marker 3 | 25.370820 | -80.160826 | Coral reef |
| LOB 120 | 25.359436 | -80.186390 | Coral reef |
| The Drop | 25.366117 | -80.137683 | Coral reef, high use by fishermen |
| Rocky Reef | 25.334617 | -80.161783 | Coral reef |
| Ball Buoy Reef | 25.318283 | -80.184317 | Coral reef |
| Elliot Key Harbor | 25.453166 | -80.196551 | Heavy boat use |
| Turkey Point Channel | 25.464164 | -80.292209 | Near power station |

liquid nitrogen vapor dry shipper for transport to the NOAA laboratory in Charleston, SC. Upon arrival, samples were stored at -80˚C until analysis.

## Sea urchin embryo development toxicity bioassay

Frozen porewater samples were thawed at 4˚C for 72 h and agitated to mix. An aliquot (5 mL) of each sample was removed to a 50 mL Falcon tube where salinity and pH were measured to ensure general water quality for the bioassay (Table 2). Salinity was adjusted (36 ±1 ppt) with the addition of either Type 1 water or dry sea salts (Tropic Marin, Wartenburg, Germany). Since ammonia can be toxic to sea urchin embryos and can confound assay results, we used a 500 μL subsample to measure total ammonia nitrogen (TAN) using a colorimetric microplate assay based on a commercial kit (Red Sea, Houston, TX, salicylate method). Ammonia standards for the assay (two-fold dilutions series, 0.13–8.0 mg/L) were generated using 100 mg/L ammonia standard (Hach, Catalog #2406549) in 36 ppt artificial seawater (ASW, Tropic Marin sea salts). Ammonia (UAN) values were calculated using a standard method [17]. Following water quality analysis, test samples (5 mL, 4 replicates) were dispensed into pre-

**Table 2. Water quality results for control and Biscayne National Park sediment porewater samples.**

| Site | Test Salinity (ppt) | pH | TAN (mg/L) | UAN (μg/L) | Color/Odor |
|---|---|---|---|---|---|
| ASW (-) control | 36 | 8.26 | 0.000 | 0.0 | clear |
| SDS, 4 mg/L (+) control | 36 | 8.27 | 0.000 | 0.0 | clear |
| *Brewster* | 37 | 7.86 | 2.968 | 103.4 | clear, slightly yellow |
| *Stiltsville A-Frame* | 36 | 7.71 | 4.190 | 104.4 | clear |
| *Stiltsville Baldwin Sessions* | 36 | 7.76 | 2.211 | 61.6 | clear |
| *Stiltsville Miami Spring Power Boat Club* | 36 | 7.58 | 3.371 | 62.7 | clear |
| *Boca Chita Harbor* | 37 | 7.68 | 3.166 | 73.7 | clear |
| *"Boiler" safety valve* | 37 | 7.56 | 1.965 | 34.9 | clear |
| **Mowry Canal** | 36 | 7.96 | 4.396 | **191.1** | tan, slightly cloudy |
| *Soldier Key* | 37 | 7.72 | 3.450 | 87.9 | clear |
| *Lirman Nursery 1* | 37 | 7.83 | 3.305 | 107.7 | clear, slightly yellow |
| *Lirman Nursery 2* | 37 | 7.81 | 3.377 | 105.3 | clear |
| Lirman Nursery 3 | 37 | 7.69 | 0.498 | 11.9 | clear |
| *University Dock* | 37 | 7.73 | 3.466 | 90.3 | clear |
| Anniversary Reef | 37 | 7.74 | 1.062 | 28.3 | clear, slightly yellow |
| *Mandalay* | 37 | 7.74 | 1.668 | 44.5 | clear, yellow, odor |
| *Alina's Reef* | 37 | 7.67 | 1.612 | 36.7 | clear |
| Marker 14 Reef | 37 | 7.54 | 1.762 | 29.9 | clear |
| Money Reef | 37 | 7.77 | 0.815 | 23.2 | clear, slightly yellow |
| Boat Basin | 36 | 7.88 | 0.064 | 2.3 | cloudy, yellow, odor |
| Marker 3 | 37 | 7.49 | 0.458 | 6.9 | clear |
| LOB 120 | 37 | 7.68 | 0.408 | 9.5 | clear, odor |
| The Drop | 37 | 7.60 | 0.587 | 11.4 | clear, odor |
| Rocky Reef | 37 | 7.73 | 0.482 | 12.6 | clear, yellow, odor |
| Ball Buoy | 37 | 7.65 | 0.560 | 12.2 | clear |
| Elliot Key Harbor | 37 | 7.85 | 0.814 | 27.7 | clear, pink, odor |
| Turkey Point Channel | 37 | 7.65 | 1.344 | 29.3 | cloudy, yellow, odor |

Ammonia (UAN) was above the lowest observable effect concentration for Mowry Canal (bolded text) and may also contribute to the toxicity at 12 additional sites (italicized site names). TAN = total ammonia nitrogen, ppt = parts per thousand, ASW = artificial seawater, negative control, SDS = sodium dodecyl sulfate, 4 mg/L, positive control.

cleaned, rinsed (5 mL ASW), 20-mL glass vials (Environmental Express, Charleston, SC) and placed in an environmentally-controlled room (26.0 +/-0.5˚C) to warm.

Toxicity of the resulting porewater was determined according to a standard method [18] using the tropical sea urchin species, *Lytechinus variegatus* [13, 19]. Gravid sea urchins were acquired from the Florida Keys (Reeftopia, Key West, FL), and held at 27˚C in a glass aquarium system containing ASW. Lighting was provided by one 1000 W, 14,000 K Hamilton Technology (Gardena, CA) metal halide bulb mounted 4 ft above the water surface and programmed to a 14h:10h light:dark cycle. Urchins were fed with organic spinach daily and organic carrots 2–3 times per week.

Urchin spawning was induced using potassium chloride (0.5 M) injections (1–3 mL) into the coelom. Eggs were extruded directly into ASW, washed three times with an equal volume of fresh ASW and enumerated on a Sedgwick-Rafter counting chamber. Sperm was collected dry by aspiration, held at 4˚C and then diluted 1:1000 in ASW (26˚C) to activate. The sperm cell concentration was determined using a hemocytometer and motility was verified. A fertilization pre-test was performed to determine the optimal sperm concentration needed to achieve a fertilization rate >95%. Embryos (50 µL, 4000/mL) were placed in glass vials containing 5 mL of test (warmed porewater) sample. Sodium dodecyl sulfate (SDS, 4 mg/L in ASW) and ASW were included as assay positive and negative controls, respectively. Vials were swirled gently to mix and the vial lids loosely attached to ensure adequate oxygenation during the course of the experiment. Embryos were incubated for 48 h at 26 ± 0.5˚C under ambient lighting on a 12h:12h light:dark cycle.

Following incubation, an equal volume of 2X zinc-formalin fixative (Anatech, Poughkeepsie, NY) (made using 2 parts 5X concentrated fixative, 2 parts 72 ppt ASW and one part 36 ppt ASW) was added to each vial. Embryo developmental stage (Fig 1) and developmental aberrations were scored: normal (four-armed pluteus), underdeveloped (two armed pluteus or prism), arrested (cell-stage, blastula or gastrula), or malformed (missing arm, missing gut,

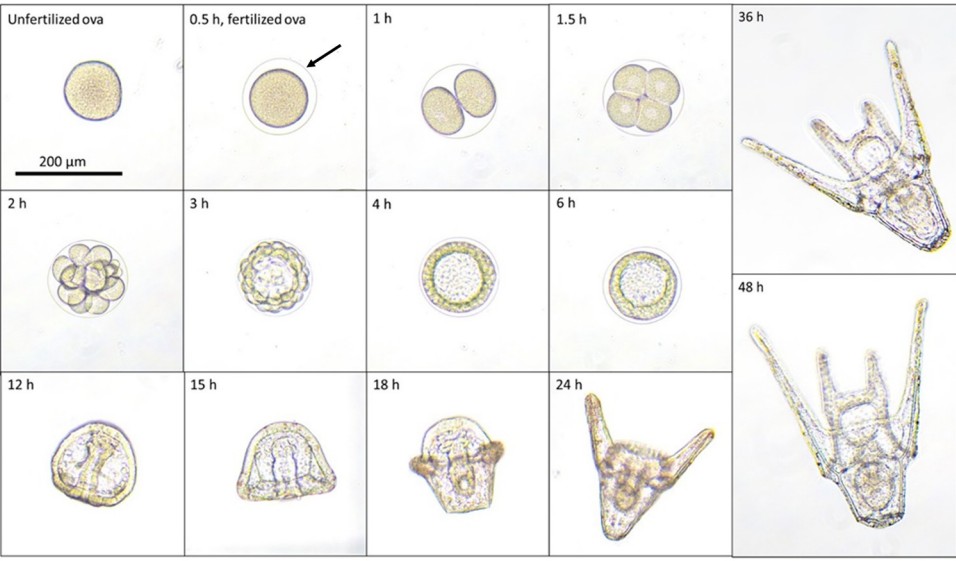

**Fig 1. The developmental stages of *Lytechinus variegatus* at 26˚C.** The time post fertilization (pf) is shown in the upper left corner of each image. Fertilization occurs within 30 min after male and female gametes are mixed together (indicated by the presence of a fertilization membrane, black arrow). Cell-stage embryos (2–64 cells) develop between 1–2 h pf. Blastulas are evident between 3–6 h pf and gastrulation is completed by 12 h pf. Prism stage is observed at 15 h pf. At 18–24 h pf, two arms appear and elongate, indicating the early pluteus developmental stage. By 36–48 h pf, the embryo has developed into a four-armed pluteus.

skeletal aberrations, etc.), with a target of 100 embryos evaluated per sample replicate. This scoring system is more refined than that described in the referenced ASTM standard method (scoring normal vs abnormal embryos, [18]), and is used as a basis for linking developmental characteristics to known toxicants. The mean percent normal embryo development and standard error for each data set were calculated and those sites with toxicity within the standard error of the mean of the positive control (percent normal = 9.75 ± 9.75, SEM) were designated toxic. The remaining data sets were subjected to a normality test (Shapiro-Wilk), followed by a one-way ANOVA with Dunnett's posttest using GraphPad Prism version 9.3.0 for Windows (GraphPad Software, La Jolla California USA, www.graphpad.com). Site impacts were determined from the statistical analysis of the percent of embryos developing normally. A Spearman's rank correlation also was computed to assess the relationship between the ammonia concentration in the sediment porewater and percent normal sea urchin embryo development.

### Ethics statement

Research ethics committee approval was not required to accomplish the goals of this study, as experiments were conducted with an unregulated invertebrate species (green sea urchin, *Lytechinus variegatus*).

## Results and discussion

### Water quality analyses for sea urchin bioassay

Water quality measurements were conducted to ensure that the conditions were suitable for the SUETOX test (Table 2). Salinity measurements were between 36–37 ppt for all test samples following adjustment and pH ranged from 7.49–7.96. Both parameters were within acceptable ranges for the test [20]. Total ammonia nitrogen ranged from 0.064 mg/L (Boat Basin) to 4.396 mg/L (Mowry Canal). We found a negative correlation (-.27) between ammonia concentration in the sediment porewater and normal embryo development (Spearman's rank test, p = 0.20). Ammonia at Mowry Canal (191 µg/L) was higher than the lowest observable effect concentration (LOEC, 149 µg/L, abnormal development) for *L. variegatus* [13], and likely contributes to toxicity at this site. We have determined mortality data from ammonia exposure for coral larvae (*Acropora palmata*, a coral species located within BISC) at 29 and 31˚C, with median lethal concentrations ($LC_{50}$) of 55 and 74 µg/L UAN, respectively (our unpublished data). Additionally, chronic ammonia exposure (16 µg/L UAN for 42 d) has been shown to reduce gonad growth in *Strongylocentrotus droebachiensis* (sea urchin) [21], that may indicate reproductive impairments for additional species such as stony corals. The US Environmental Protection Agency [22] seawater chronic (4-day average) criterion for UAN is <35 µg/L. While our data represent a single time point, it is possible that ammonia may impact 12 additional sites (Table 2, site names in italicized text), including two coral nursery sites (Lirman Nursery 1 and 2) and Alina's Reef. We recommend routine testing of seawater samples (TAN, pH, salinity and temperature) for monitoring UAN, to better gauge potential impacts from ammonia within BISC.

### Sea urchin embryo development toxicity assay

The SUETOX assay was used to evaluate the effects of the various BISC sediment porewater treatments (raw data, S1 Table). The percent normal embryo development for all control and test samples is shown in Fig 2. As expected, 90.5% of the embryos incubated in ASW developed normally and embryos incubated in the SDS positive control exhibited delayed development

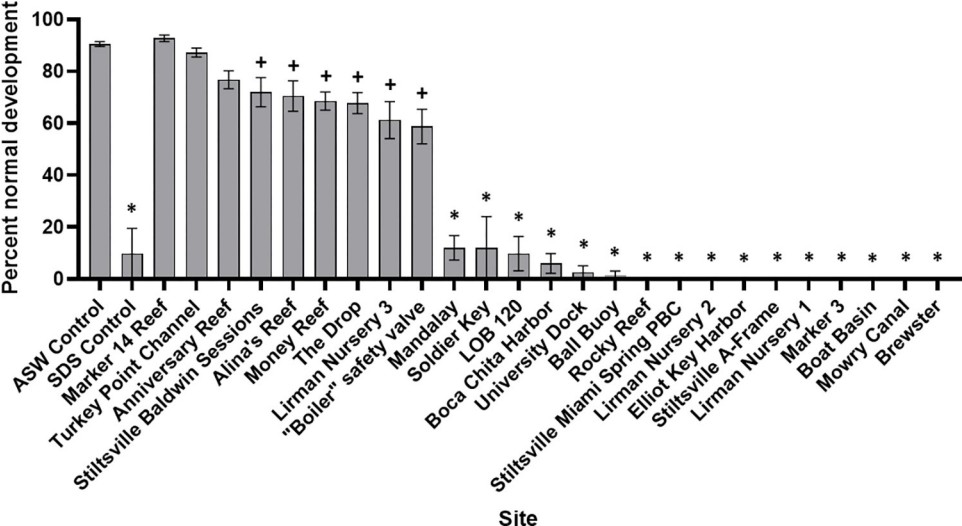

**Fig 2. Percent normal results of the sea urchin embryo development toxicity assay.** Three of the 25 Biscayne National Park sites were not significantly different from the artificial seawater (ASW) control. Six sites were impacted (significantly different from the ASW control at p < 0.0001, designated with '+' symbol). The remaining 16 sites were severely impacted (toxic) with mean percent normal development less than 20% (designated by '*'). SDS treatment (positive control) = 4 mg/L sodium dodecyl sulfate in ASW.

after 48 h (9.8% normal embryos). Fig 3 shows the relative proximities of percent normally developed embryos at each BISC site. No toxicity was observed at three of the 25 BISC sites (mean percent normal embryos in parentheses): Anniversary Reef (76.8%), Marker 14 Reef (92.8%) and Turkey Point Channel (86.8%) (Fig 3, green triangles). These sites had a range of UAN between 28.3–29.9 µg/L and pH from 7.54 to 7.74 (Table 2). Six sites were impacted (mean percent normal significantly different (ANOVA, p < 0.0001) from the ASW control): Stiltsville Baldwin Sessions (72.0% normal) "Boiler" Safety Valve (58.9% normal), Alina's Reef (70.5% normal), Money Reef (68.5% normal), The Drop (68.6% normal) and Lirman Nursery 3 (61.3% normal) (Fig 3, yellow triangles). Ammonia in porewater samples from these sites ranged from 11.4–61.6 µg/L. It is possible that ammonia (>29.9 µg/L) may contribute to toxicity at Stiltsville Baldwin Sessions, Boiler Safety Valve and Alina's Reef.

The remaining 16 sites were toxic to sea urchin embryos (Fig 3, red triangles). Of these, ten sites had UAN levels above 30 µg/L (i. e., above the range of UAN levels at non-toxic sites) and one (Mowry Canal) was as high as 191 µg/L, in excess of the *L. variegatus* LOEC of 148.9 µg/L. As mentioned above, the UAN at Mowry Canal likely contributes to the observed toxicity at that site. Mowry Canal drains an area proximal to Homestead FL, and could be impacted by multiple stressors including Homestead Air Reserve Base, Homestead Hospital, and high density residential, commercial and agricultural use areas (the latter three, likely sources of ammonia from fertilizer applications). Chronic UAN levels above the US EPA [22] criterion of 35 µg/L could have negatively affected sea urchin embryo development at many of these other sites. In addition to Mowry Canal, 9 of the 16 severely impacted sites had UAN levels above this criterion and that also exceeded levels at non-toxic sites (Table 2). However, it cannot be determined from this initial screen if the impacts are related to UAN alone, or in addition to other stressors. We did not observe an impact of sediment type on embryo development (S1 Fig). We also note that embryos incubated in sediment porewater from three sites (Ball Buoy, Boat Basin and Mowry Canal) likely had substantial mortality since less than 100 embryos could be found in each replicate sample vial (out of ~200 embryos initially added per vial).

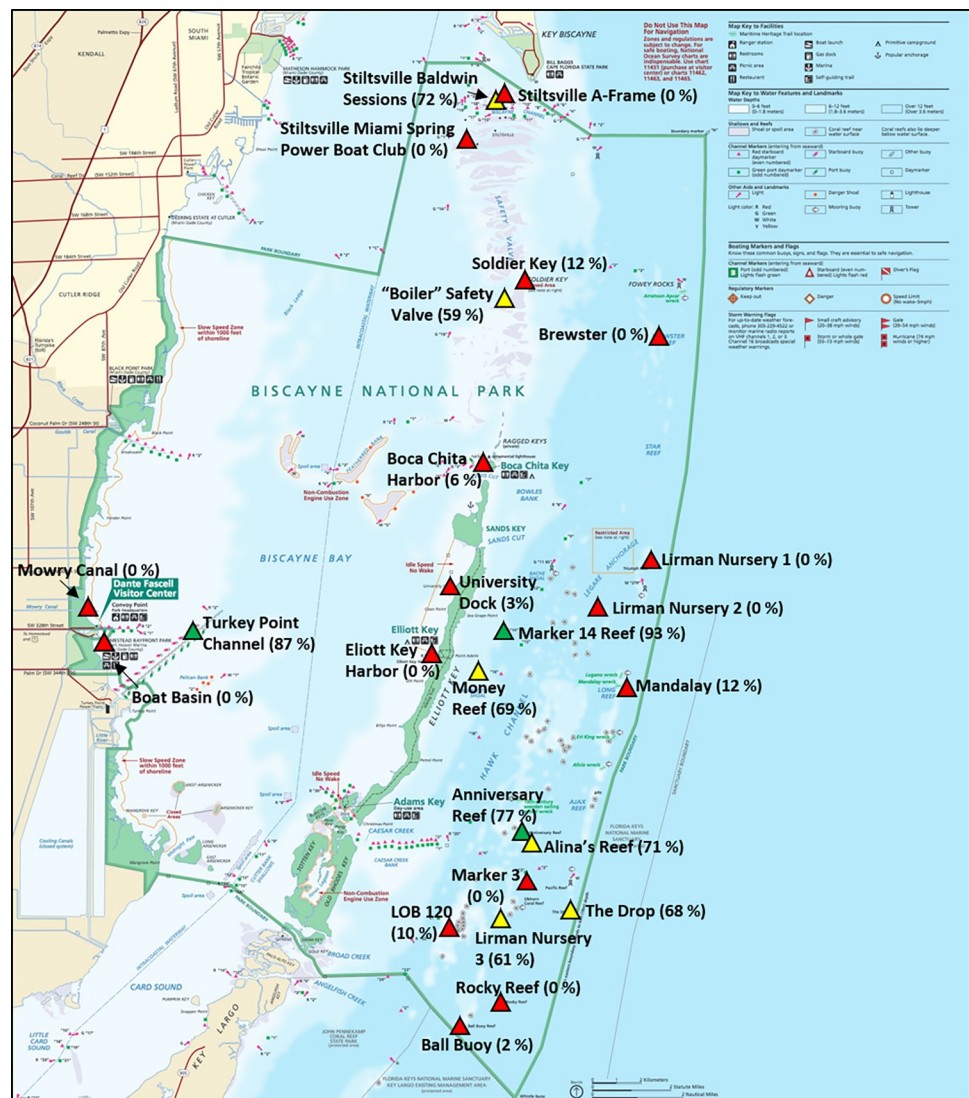

**Fig 3. Map of Biscayne National Park sediment sampling sites with toxicity assay results.** Non-impacted sites = green triangles; impacted sites = yellow triangles and toxic sites = red triangles. Mean percent normal sea urchin embryo development is in parentheses following the site name. Park boundary = solid green line. (Map source: National Park Service, public domain).

Interruptions in the sea urchin embryo developmental program were observed at 22 out of 25 BISC sites sampled (Figs 4 and 5). We observed that sea urchin embryo underdevelopment was the dominant phenotype following incubation in porewater from most BISC sites. Most sites associated with regular boat activity, such as harbors or moorings (e.g., Boca Chita Harbor, University Dock, Boat Basin, Ball Buoy, Elliot Key Harbor, Alina's Reef, Mandalay, The Drop, Rocky Reef and Ball Buoy) had significant urchin developmental impacts, possibly due to release of antifoulant or hydrocarbon products associated with boating activity. Both biodegraded crude oil [23] and boat antifoulants such as diuron, irgarol, copper, tributyltin (TBT) and triphenyltin (TPT) have been shown to delay development of sea urchin embryos [24–28]. While the manufacture and use of TBT and TPT have been banned in the United States for more than 20 years, both compounds are still used in several countries around the world, including nearby Caribbean nations. Additionally, TBT does not degrade quickly and may

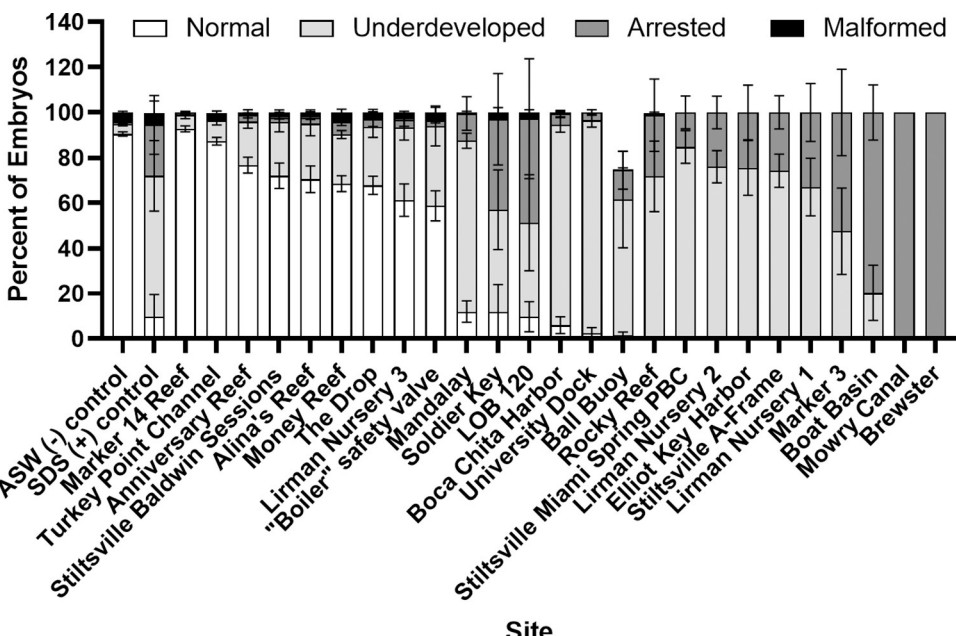

**Fig 4. Percent sea urchin embryo development.** The percent of normal, underdeveloped, arrested or malformed urchin embryos is presented for each of the Biscayne National Park sites. ASW = artificial seawater (negative control); SDS = 4 mg/L sodium dodecyl sulfate in ASW (positive control).

remain in anaerobic marine sediments for decades [29]. The last comprehensive environmental assessment of BISC [30] revealed the presence of organotin compounds in marine sediments proximal to Mowry Canal, Boat Basin and Turkey Point Channel (TBT = 1.56, 22.10 and 0.35 ng Sn/g dry weight, respectively). The tin analytes are particularly toxic, with a median effect concentration ($EC_{50}$) range of 0.36–0.73 µg/L for *L. variegatus* [27]. Perina et al. [27] postulate that developmental effects could be due to TBT or TPT interference in calcium homeostasis, which, if true, could negatively affect other marine organisms with calcium carbonate skeletons, such as stony corals.

Metal exposures (e.g., copper, lead, zinc) can delay sea urchin embryo development [31] and differences in embryo aberrations resulting from metals exposure are concentration-dependent, with arrested development resulting from higher concentrations [28, 32]. Therefore, we acknowledge the possibility that metal contamination could play a role in toxicity in BISC, particularly for those sites resulting in high percentages of arrested embryos (Fig 4). Copper and zinc contamination previously have been documented on the western shoreline of BISC, with a high probable Effect Index for these metals co-located with the Boat Basin and Mowry Canal sites in the current study [33]. Cantillo and Lauenstein [30] also measured several metals near the mouths of canals, and sites proximal to Mowry Canal and Boat Basin had total trace metals (including Cu, Hg and Pb) of 75 and 244 µg/g dry weight. Arrested development also can result from exposure to insecticides (e.g., diazinon, carbaryl, pirimicarb) [34], with likely sources of these contaminants coming from nearby residential landscaping and agricultural use areas. Evidence of pesticide contamination (diazinon, atrazine, DDT, etc.) from canals in BISC has been documented [30].

In addition to hydrocarbon, antifoulant and pesticide releases, human recreational activities and sewage input from urban centers are a source of contaminants of emerging concern. Personal care products such as sunscreens are washed from the body into the water during swimming or into sewage treatment plants or septic systems during bathing, and contain additives

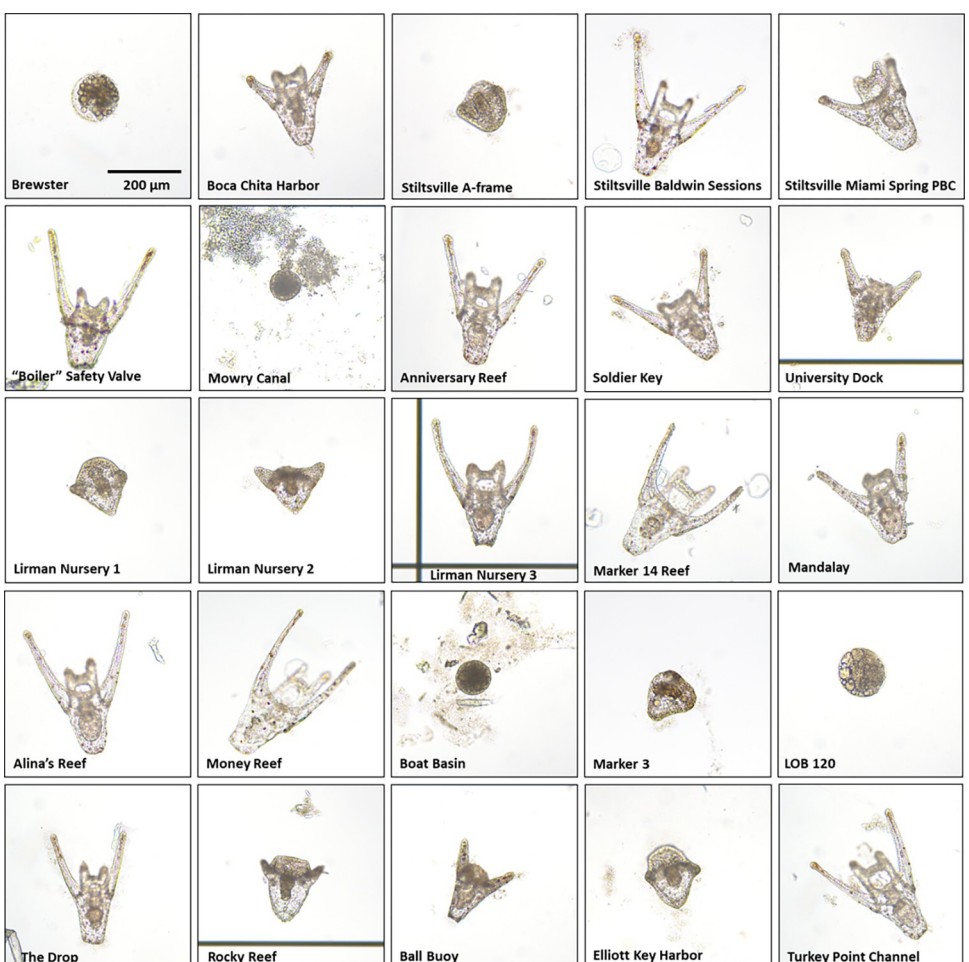

**Fig 5. Representative sea urchin embryo images for each Biscayne National Park site.** Magnification = 100x. All images are to scale.

(UV filters, preservatives, etc.) not removed by treatment and that impact sea urchin [35–37] and coral embryo development [38]. Release of wastewaters also can deposit pharmaceuticals (e.g., antibiotics, analgesics, etc.) and detergents into the water column, many that have been engineered to resist biotic and abiotic degradation, and which have a high propensity for binding to biota or sediments [39]. Many of these types of compounds are toxic to sea urchin embryos and other marine organisms [40–43]. It is likely BISC receives some sewage input from leaky septic systems and impeded water flow resulting from habitat fragmentation (canal construction) allows higher retention of these types of compounds in the sediment reservoir [44]. A 2017 survey of select sites within BISC (including Anniversary Reef and Boca Chita Harbor in the current study) showed that concentrations of select UV filters (avobenzone, oxybenzone, octocrylene and dioxybenzone) were detected from water column samples, but below levels known to cause acute effects in marine organisms (maximum concentration range 14–125 ng/L) [45]. The authors stress that it would be beneficial to determine the frequency of UV filters in BISC in a more formalized (vs opportunistic) study.

We did not observe significant embryonic malformations (e.g., skeletal aberrations, missing arms, missing gut, absence of pigment cells) in any sample (Fig 4). This was unexpected, particularly with respect to the evident toxicity at most sites. Contaminants documented to cause

malformations in sea urchin embryos include glyphosate [46], PCBs and PAHs [40, 47], low concentrations of metals or pesticides [28, 34], nanoparticles [48] and antibiotics [49]. PCB and PAH contamination have been documented at both Mowry Canal and Boat Basin [30]. Data from the NOAA National Status and Trends website also indicates presence of several sediment contaminants (metals, PAHs, PCBs, and biocides such as TBT) from a single site on the north-central Park border (Gould's Canal, 2001–2009). Since sampling and chemical analyses for these studies were last performed in 2009, it is not known if these compounds persist in BISC sediments.

The SUETOX assay is an underused tool for determining biological impacts from anthropogenic stressors in sensitive marine ecosystems. This is surprising, since it is a relatively inexpensive method for performing initial toxicity screens of marine sediments and/or waters. The SUETOX assay results can be used to focus costly analytical chemistry screens to areas demonstrating significant impacts, and also may be a more sensitive indicator of environmental contamination, depending on the sensitivity of the analytical methods used. Identifying the sources of the toxicants within BISC (e.g., Toxicity Identification Evaluations (TIE) [50] and/ or contaminant chemistry analyses) could pinpoint possible sources of toxicants affecting protected resources in nearshore habitats such as BISC and guide management actions. With increasing attempts to reintroduce corals to degraded habitats, the SUETOX assay could be used to screen historical reef habitats for restoration suitability.

## Conclusions

We observed no toxicity at three BISC sites using the SUETOX test: Anniversary Reef, Marker 14 Reef and Turkey Point Channel. Six sites were impacted (significantly different from the ASW control, ANOVA, $p < 0.0001$) and the remaining 16 sites were toxic to sea urchin embryos. Mowry Canal toxicity may have been due, at least in part, to high levels of ammonia (191.1 µg/L) that exceeded the *L. variegatus* LOEC of 148.9 µg/L. Levels of ammonia that were in excess of the US EPA chronic criterion of 35 µg/L [22] and that also exceeded levels at nontoxic sites may have contributed to toxicity at twelve additional sites. Due to moderate to high levels of ammonia in many BISC porewater samples, routine monitoring for ammonia (TAN, pH, salinity and temperature measurements) is recommended. The results reveal that discrete areas of BISC have highly toxic sediments; the presence of sediment contaminants could negatively impact reproduction, growth and population density of other invertebrates, such as stony corals. Results of the SUETOX test using BISC sediment may prompt changes in policy and management decisions within the Park and stress the need for ongoing monitoring of toxicity conditions. Since the last comprehensive environmental assessment of BISC was performed in 1999, further investigation into the causes and sources of the toxicity at BISC is needed.

## Supporting information

**S1 Table. Raw data for the sea urchin embryo development test.**
(XLSX)

**S1 Fig. Comparison of sediment type and sea urchin embryo development.** Examples of sediment types found during the study included 1) high silt with dark organic matter (panels A and D), 2) sand with some organic matter (panels B and E), and 3) coarse sand and coral rubble (panels C and F). Normal embryo development was observed in porewater from samples in panels A-C. Embryo development was significantly impacted following incubation in porewater from sediment samples D-F.
(TIF)

## Acknowledgments

We would like to thank three NOAA scientists, Dr. Jeff Hyland, Dr. Dave Whitall and Dr. Paul Pennington, and two journal reviewers for critical comments which greatly improved the manuscript. We appreciate assistance with statistical analyses from Dr. John Fauth (University of Central Florida). We also would like to thank Michael Hoffman (National Park Service, Biscayne) for sampling assistance.

## Author Contributions

**Conceptualization:** Catherine Anna Toline, Cheryl M. Woodley.

**Data curation:** Lisa A. May.

**Formal analysis:** Lisa A. May.

**Funding acquisition:** Catherine Anna Toline, Cheryl M. Woodley.

**Investigation:** Lisa A. May, Elizabeth Murphy McDonald, Ronald T. Kothera, Zachary J. Moffitt.

**Methodology:** Lisa A. May.

**Project administration:** Cheryl M. Woodley.

**Resources:** Catherine Anna Toline, Vanessa McDonough, Zachary J. Moffitt, Carl V. Miller, Cheryl M. Woodley.

**Supervision:** Lisa A. May, Cheryl M. Woodley.

**Validation:** Lisa A. May.

**Visualization:** Lisa A. May.

**Writing – original draft:** Lisa A. May.

**Writing – review & editing:** Catherine Anna Toline, Vanessa McDonough, Cheryl M. Woodley.

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
