## [Decision Letter · Decision Letter 0]

1 Sep 2022

PONE-D-22-17111Assessment of Sediment Porewater Toxicity in Biscayne National ParkPLOS ONE

Dear Dr. May,

Thank you for submitting your manuscript to PLOS ONE. After careful consideration, we feel that it has merit but does not fully meet PLOS ONE’s publication criteria as it currently stands. Therefore, we invite you to submit a revised version of the manuscript that addresses the points raised during the review process.

We look forward to receiving your revised manuscript.

Kind regards,

Xiaoshan Zhu, Ph.D.

Academic Editor

PLOS ONE

Journal Requirements:

3. We note that Figure 3 in your submission contain satellite images which may be copyrighted. All PLOS content is published under the Creative Commons Attribution License (CC BY 4.0), which means that the manuscript, images, and Supporting Information files will be freely available online, and any third party is permitted to access, download, copy, distribute, and use these materials in any way, even commercially, with proper attribution. For these reasons, we cannot publish previously copyrighted maps or satellite images created using proprietary data, such as Google software (Google Maps, Street View, and Earth). For more information, see our copyright guidelines: http://journals.plos.org/plosone/s/licenses-and-copyright.

a. You may seek permission from the original copyright holder of Figure 3 to publish the content specifically under the CC BY 4.0 license. 

Reviewers' comments:

Reviewer's Responses to Questions

**Comments to the Author**

1. Is the manuscript technically sound, and do the data support the conclusions?

Reviewer #1: Yes

Reviewer #2: Partly

2. Has the statistical analysis been performed appropriately and rigorously? 

Reviewer #1: Yes

Reviewer #2: Yes

3. Have the authors made all data underlying the findings in their manuscript fully available?

Reviewer #1: Yes

Reviewer #2: Yes

4. Is the manuscript presented in an intelligible fashion and written in standard English?

Reviewer #1: Yes

Reviewer #2: Yes

5. Review Comments to the Author

Reviewer #1: 1. Generally, the ionic composition and redox condition of porewater are different from normal seawater. Whether sediment porewater itself is suitable for sea urchin embryo development when salinity and ammonia were not considered ?

2. Please demonstrate the type of sediment if any. Whether sea urchin embryo development in porewater is related to sediment type?

3. Correlation analysis between embryo development and ammonia is needed.

4. Please state the function of SDS Controls in the section of Materials and methods.

5. Toxicity screening is needed for further works.

Reviewer #2: Generally, the article is easy to follow. The authors collected sediment samples and performed the SUETOX test on the extracted porewater. The article discusses the toxicity of sediments and points out the need to identify the source. This is a general investigation and the following questions need to be carefully addressed.

1. The title is not specific enough. The target biota should at least be involved.

2. Why didn’t the authors analyze the contaminants in the target sediments? It would be much reasonable to at least show which contaminants were mainly in the target sediments.

L114 Do authors have any data to support the expectation?

L117 What’s the full name of NPS?

L122 The “x” should be a multiplication sign.

L123, 125 The hyphen should be a minus.

L143 No need to set all the parameters for Mowry Canal to bold.

L153 Should be “fed with”?

L232 The red labels were vague.

6. PLOS authors have the option to publish the peer review history of their article (what does this mean?). If published, this will include your full peer review and any attached files.

Reviewer #1: No

Reviewer #2: **Yes: **Ciara Chun Chen

---

## [Author Response · Author response to Decision Letter 0]

28 Oct 2022

All requested revisions have been addressed in the 'Revised Manuscript with Track Changes' and in the 'Response to Reviewers' documents. A copy of our responses is listed below.

Response to Reviewers

Journal Requirements:

Authors’ response: We have corrected formatting for file naming, author affiliations and references (brackets instead of parentheses).

Authors’ response: We have added an ethics statement to the ‘Materials and methods’ section to comply with this requirement.

3. We note that Figure 3 in your submission contain satellite images which may be copyrighted. All PLOS content is published under the Creative Commons Attribution License (CC BY 4.0), which means that the manuscript, images, and Supporting Information files will be freely available online, and any third party is permitted to access, download, copy, distribute, and use these materials in any way, even commercially, with proper attribution. For these reasons, we cannot publish previously copyrighted maps or satellite images created using proprietary data, such as Google software (Google Maps, Street View, and Earth). For more information, see our copyright guidelines: http://journals.plos.org/plosone/s/licenses-and-copyright.

a. You may seek permission from the original copyright holder of Figure 3 to publish the content specifically under the CC BY 4.0 license. 

Authors’ response: We have changed the Fig 3 map of Biscayne National Park to one that is in the public domain (source, National Park Service).

Reviewers' comments:

Reviewer's Responses to Questions

Comments to the Author

1. Is the manuscript technically sound, and do the data support the conclusions?

Reviewer #1: Yes

Reviewer #2: Partly

2. Has the statistical analysis been performed appropriately and rigorously?

Reviewer #1: Yes

Reviewer #2: Yes

3. Have the authors made all data underlying the findings in their manuscript fully available?

Reviewer #1: Yes

Reviewer #2: Yes

4. Is the manuscript presented in an intelligible fashion and written in standard English?

Reviewer #1: Yes

Reviewer #2: Yes

5. Review Comments to the Author

Reviewer #1: 1. Generally, the ionic composition and redox condition of porewater are different from normal seawater. Whether sediment porewater itself is suitable for sea urchin embryo development when salinity and ammonia were not considered ?

Authors’ response: The authors would like to thank Reviewer 1 for the critical comments. The sea urchin embryo development test is a standard method (ASTM and US EPA) and has been used by other research groups to assess the toxicity of marine porewaters. Biscayne National Park is a habitat for Lytechinus variegatus, so this is a logical species to select for this toxicity assay. Normal sea urchin embryo development was reported for three sites in this study and across sediment types (see #2 below), indicating that the ionic and redox composition of porewater from various sediment types is suitable for urchin embryo development. Both salinity and ammonia were measured in all porewater samples prior to the test. Salinity was adjusted to within 1 ppt of the artificial seawater control, a range which does not negatively affect L. variegatus embryo development. Total ammonia nitrogen was measured in each sample, however it is not easily removed without potentially altering the chemical characteristics of the porewater (and altering any potential chemical contaminants). We therefore decided to report ammonia and state the caveat that ammonia could contribute to the toxicity at a site where it was above the reported EC50 or US EPA chronic criterion. References are given for both criteria. We have additionally performed a Spearman ranked correlation test of sediment porewater ammonia concentration and percent normal embryo development, per comments below. We found no correlation between these parameters and have reported these results in the revised manuscript.

2. Please demonstrate the type of sediment if any. Whether sea urchin embryo development in porewater is related to sediment type?

Authors’ response: Sediment type varied across the sample sets, from a relatively high silt and organic matter content (noted in images and visual inspection) to more coarse coral rubble, typically found in and around stony coral reef areas. We observed normal sea urchin embryo development across the range of sediment types, indicating that this parameter had no impact on the assay. We have included a supplemental figure with example images from six sites, two of which were high in silt content and darker in color (possible higher organic matter content), two which were sand and silt with some organic matter and two which were of the consistency of coarse sand, shell and coral rubble. Each sediment type in the figure had both normal embryo development (panels A -C) and impacted embryo development (panels D-F). 

3. Correlation analysis between embryo development and ammonia is needed.

Authors’ response: A Spearman’s rank correlation (data were not normally distributed) was computed to assess the relationship between ammonia concentration in the sediment porewater and sea urchin percent normal embryo development. There was a negative correlation between the two variables, r(23) = -.27, p = 0.20. We have reported this in the Results and discussion section of the revised manuscript.

4. Please state the function of SDS Controls in the section of Materials and methods.

Authors’ response: We stated that the sodium dodecyl sulfate (SDS) functions as the reference toxicant (assay positive control, Materials and methods, lines 168-169 of the original manuscript).

5. Toxicity screening is needed for further works.

Authors’ response: The authors agree. Future work will incorporate a toxicity identification evaluation (TIE) to narrow the class of target analytes, followed by targeted analytical chemistry. This will help inform Park managers of potential risks to Park resources and guide management actions to reduce those risks.

Reviewer #2: Generally, the article is easy to follow. The authors collected sediment samples and performed the SUETOX test on the extracted porewater. The article discusses the toxicity of sediments and points out the need to identify the source. This is a general investigation and the following questions need to be carefully addressed.

1. The title is not specific enough. The target biota should at least be involved.

Authors’ response: The authors would like to thank Reviewer 2 for all critical comments. We have revised the title to be more specific and have included the target test organism.

2. Why didn’t the authors analyze the contaminants in the target sediments? It would be much reasonable to at least show which contaminants were mainly in the target sediments.

Authors’ response: This study was undertaken to avoid costs associated with unnecessary and/or non-targeted analytical chemistry analyses. As mentioned above (Reviewer 1, comment #5), future work will entail Level 1 screening using a toxicity identification evaluation (TIE) and follow up chemistry analyses to determine the nature of the toxicity.

L114 Do authors have any data to support the expectation?

Authors’ response: Site selection was performed by National Park Service personnel and based upon visual observations for those areas of high recreational use. Marinas and boat harbors were selected to see if those sites exhibited toxicity related to boat use. Coral reef and coral nursery habitats were selected since those are a protected resource within the Park jurisdiction, and the Park is tasked with protecting those resources. 

L117 What’s the full name of NPS?

Authors’ response: We have changed this abbreviation to ‘National Park Service’

L122 The “x” should be a multiplication sign.

Authors’ response: We have changed the ‘x’ to ‘by’ since this is the dimension of the storage vessel.

L123, 125 The hyphen should be a minus.

Authors’ response: We inserted a minus symbol, however in this font, this symbol has a dual role as a hyphen.

L143 No need to set all the parameters for Mowry Canal to bold.

Authors’ response: The authors agree and have eliminated bold text for all parameters except the ammonia (UAN) concentration.

L153 Should be “fed with”?

Authors’ response: We have added ‘with’ to the sentence indicated.

L232 The red labels were vague.

Authors’ response: The authors agree. We have revised Fig 3 and have changed the font color to black for all site names to address this issue.

6. PLOS authors have the option to publish the peer review history of their article (what does this mean?). If published, this will include your full peer review and any attached files.

Do you want your identity to be public for this peer review? For information about this choice, including consent withdrawal, please see our Privacy Policy.

Reviewer #1: No

Reviewer #2: Yes: Ciara Chun Chen

---

## [Decision Letter · Decision Letter 1]

22 Nov 2022

Assessment of Sediment Porewater Toxicity in Biscayne National Park with Sea Urchin (Lytechinus variegatus) embryos

PONE-D-22-17111R1

Dear Dr. May,

We’re pleased to inform you that your manuscript has been judged scientifically suitable for publication and will be formally accepted for publication once it meets all outstanding technical requirements.

Kind regards,

Xiaoshan Zhu, Ph.D.

Academic Editor

PLOS ONE

Additional Editor Comments (optional):

Reviewers' comments:

Reviewer's Responses to Questions

**Comments to the Author**

1. If the authors have adequately addressed your comments raised in a previous round of review and you feel that this manuscript is now acceptable for publication, you may indicate that here to bypass the “Comments to the Author” section, enter your conflict of interest statement in the “Confidential to Editor” section, and submit your "Accept" recommendation.

Reviewer #1: All comments have been addressed

Reviewer #2: All comments have been addressed

2. Is the manuscript technically sound, and do the data support the conclusions?

Reviewer #1: Yes

Reviewer #2: Yes

3. Has the statistical analysis been performed appropriately and rigorously? 

Reviewer #1: Yes

Reviewer #2: Yes

4. Have the authors made all data underlying the findings in their manuscript fully available?

Reviewer #1: Yes

Reviewer #2: Yes

5. Is the manuscript presented in an intelligible fashion and written in standard English?

Reviewer #1: Yes

Reviewer #2: Yes

6. Review Comments to the Author

Reviewer #1: (No Response)

Reviewer #2: (No Response)

7. PLOS authors have the option to publish the peer review history of their article (what does this mean?). If published, this will include your full peer review and any attached files.

Reviewer #1: No

Reviewer #2: No

---

## [Editor Report · Acceptance letter]

28 Nov 2022

PONE-D-22-17111R1 

Assessment of Sediment Porewater Toxicity in Biscayne National Park with Sea Urchin (*Lytechinus variegatus*) Embryos 

Dear Dr. May:

I'm pleased to inform you that your manuscript has been deemed suitable for publication in PLOS ONE. Congratulations! Your manuscript is now with our production department. 

Kind regards, 

on behalf of

Dr. Xiaoshan Zhu 

Academic Editor

PLOS ONE